# NeuSymEA: Neuro-symbolic Entity Alignment via Variational Inference

**Shengyuan Chen**
Department of Computing
The Hong Kong Polytechnic University
Hung Hom, Hong Kong SAR
sheng-yuan.chen@polyu.edu.hk

**Zheng Yuan**
Department of Computing
The Hong Kong Polytechnic University
Hung Hom, Hong Kong SAR
yzheng.yuan@connect.polyu.hk

**Qinggang Zhang**[*]
Department of Computing
The Hong Kong Polytechnic University
Hung Hom, Hong Kong SAR
qinggangg.zhang@connect.polyu.hk

**Wen Hua**
Department of Computing
The Hong Kong Polytechnic University
Hung Hom, Hong Kong SAR
wency.hua@polyu.edu.hk

**Jiannong Cao**
Department of Computing
The Hong Kong Polytechnic University
Hung Hom, Hong Kong SAR
csjcao@comp.polyu.edu.hk

**Xiao Huang**
Department of Computing
The Hong Kong Polytechnic University
Hung Hom, Hong Kong SAR
xiaohuang@comp.polyu.edu.hk

## Abstract

Entity alignment (EA) aims to merge two knowledge graphs (KGs) by identifying equivalent entity pairs. Existing methods can be categorized into symbolic and neural models. Symbolic models, while precise, struggle with substructure heterogeneity and sparsity, whereas neural models, although effective, generally lack interpretability and cannot handle uncertainty. We propose NeuSymEA, a unified neuro-symbolic reasoning framework that combines the strengths of both methods to fully exploit the cross-KG structural pattern for robust entity alignment. NeuSymEA models the joint probability of all possible pairs' truth scores in a Markov random field, regulated by a set of rules, and optimizes it with the variational EM algorithm. In the E-step, a neural model parameterizes the truth score distributions and infers missing alignments. In the M-step, the rule weights are updated based on the observed and inferred alignments, handling uncertainty. We introduce an efficient symbolic inference engine driven by logic deduction, enabling reasoning with extended rule lengths. NeuSymEA achieves a significant 7.6% hit@1 improvement on DBP15K$_{\text{ZH-EN}}$ compared with strong baselines and demonstrates robustness in low-resource settings, achieving 73.7% hit@1 accuracy on DBP15K$_{\text{FR-EN}}$ with only 1% pairs as seed alignments. Codes are released at https://github.com/chensyCN/NeuSymEA-NeurIPS25.

## 1 Introduction

Knowledge graphs (KGs) are crucial for organizing structured knowledge about entities and their relationships, enhancing search capabilities across various applications. They are widely used in

---

[*]Correpsponding author

39th Conference on Neural Information Processing Systems (NeurIPS 2025).

question-answering systems [1, 2], recommendation systems [3, 4], SQL generation [5, 6], natural language processing [7–9], etc.. Despite their utility, real-world KGs often face issues like incompleteness, domain specificity, and language constraints, which hinder their effectiveness in cross-disciplinary or multilingual contexts [10, 11]. To address these issues, entity alignment (EA) aims to merge disparate KGs into a unified, comprehensive knowledge base by identifying and linking equivalent entities across different KGs. For example, aligning entities between a biomedical KG and a pharmaceutical KG allows for mining cross-discipline relationships through the aligned entities, such as identifying the same drugs and their effects on different diseases to enhance drug repurposing efforts. This alignment enables more nuanced exploration and interrogation of interconnected data, providing richer insights into how entities function across multiple domains.

Entity alignment models aim to determine the equivalence of two entities by assessing their alignment probability. Existing methods can be broadly categorized into symbolic models and neural models. Symbolic models [12–14] provide interpretable and precise inference by mining ground rules, but they struggle with aligning low-degree entities, especially those without aligned neighbors. In such cases, the lack of supporting rules leads to low recall. Conversely, neural models, such as translation models [15, 16] and graph convolutional networks (GCNs) [17–23], excel in recalling similar entities by embedding them in a continuous space, yet they often fail to distinguish entities with similar representations, causing a drop in precision as the entity pool grows. Neuro-symbolic models aim to combine the strengths of both approaches, offering robust reasoning ability for entity alignment in challenging scenarios.

However, neuro-symbolic reasoning in entity alignment (EA) faces several challenges. First, combining symbolic and neural models into a unified framework is suboptimal due to the difficulty in aligning their objectives. Current approaches either use neural models as auxiliary modules for symbolic models to measure entity similarity [14] or employ symbolic models to refine pseudo-labels [24, 25]. Second, in EA task, the search space for rules is large. Deriving ground rules from both intra-KG and inter-KG structural patterns results in an exponentially growing search space as rule length increases, complicating efficient rule weight estimation and inference. Finally, generating interpretations for EA remains underexplored. Effective interpretations should not only generate supporting rules but also quantify their confidence through rule weights.

To overcome these challenges, we propose NeuSymEA, a neuro-symbolic framework that combines the strengths of both symbolic and neural models. NeuSymEA models the joint probability of truth score assignment for all possible entity pairs using a Markov random field, regulated by a set of weighted rules. This joint probability is optimized via a variational EM algorithm. During the E-step, a neural model parameterizes the truth scores and infers the missing alignments. In the M-step, the rule weights are updated based on both observed and inferred alignments. To leverage long rules without suffering from the exponential search space, we employ logic deduction to decompose rules of any length into a set of unit-length sub-rules. This allows for efficient inference and weight updates for long rules. After training, the missing alignments are jointly inferred by both components. Additionally, we introduce an explainer to enhance interpretability. By reversing the rule decomposition process, we extract long rules as explicit supporting evidence for alignments and recover rule weights as quantified confidence scores. Our contributions are summarized as below:

- **A principled neuro-symbolic reasoning framework via variational EM:** While variational EM has been utilized in KG completion tasks [26, 27], adapting it directly to the EA task is nontrivial because they only consider single-KG structures. We bridge this gap by formulating truth scores and weighted cross-KG rules, and modeling the joint probability of the truth scores in a Markov random field regulated by the weighted cross-KG rules.

- **Efficient optimization via logical decomposition:** We introduce a logic deduction mechanism that decomposes long rules into shorter ones, significantly reducing the complexity of rule inference and enabling efficient reasoning over large knowledge graphs.

- **Interpretable inference:** The explainer utilizes learned rules to generate support paths for interpreting both aligned and misaligned pairs. It offers two modes: **(1) Hard-anchor mode**—generates supporting paths from prealigned anchor pairs; and **(2) Soft-anchor mode**—incorporates inferred anchor pairs for more informative interpretation.

- **Empirical validation and superior results:** NeuSymEA demonstrates state-of-the-art performance on benchmark datasets, delivering robust alignment accuracy and rule-based interpretations.

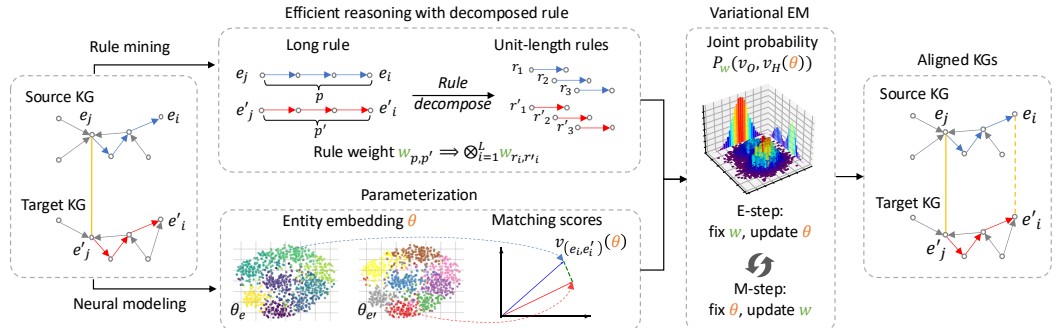

Figure 1: Framework illustration of NeuSymEA. The yellow solid line represents a pre-aligned entity pair. The symbolic model computes the path-level matching probability of entity pairs by mining rules with learned weights. The neural model learns embeddings and calculates entity-level matching scores. NeuSymEA models their agreement using a joint probability distribution over observed pairs and parameterized truth scores for hidden pairs, optimizing through a variational EM algorithm.

## 2 Preliminaries

### 2.1 Problem statement

A knowledge graph $\mathcal{G}$ comprises a set of entities $\mathcal{E}$, a set of relations $\mathcal{R}$, and a set of relation triples $\mathcal{T}$ where each triple $(e_i, r_k, e_j) \in \mathcal{T}$ represents a directional relationship between its head entity and tail entity. Given two KGs $\mathcal{G} = \{\mathcal{E}, \mathcal{R}, \mathcal{T}\}$, $\mathcal{G}' = \{\mathcal{E}', \mathcal{R}', \mathcal{T}'\}$, and a set of observed aligned entity pairs $\mathcal{O} = \{(e_i, e_i')|e_i \in \mathcal{E}, e_i' \in \mathcal{E}'\}_{i=1}^{n}$, the goal of entity alignment is to infer the missing alignments by reasoning with the existing alignments. This problem can be formulated in a probabilistic way: each pair $(e, e'), e \in \mathcal{E}, e' \in \mathcal{E}'$ is associated with a binary indicator variable $\boldsymbol{v}_{(e,e')}$. $\boldsymbol{v}_{(e,e')} = 1$ means $(e, e')$ is an aligned pair, and $\boldsymbol{v}_{(e,e')} = 0$ otherwise. Given some observed alignments $\boldsymbol{v}_O = \{\boldsymbol{v}_{(e,e')} = 1\}_{(e,e')\in\mathcal{O}}$, we aim to predict the labels of the remaining hidden entity pairs $\mathcal{H} = \mathcal{E} \times \mathcal{E}' \backslash \mathcal{O}$, i.e., $\boldsymbol{v}_H = \{\boldsymbol{v}_{(e,e')}\}_{(e,e')\in\mathcal{H}}$.

### 2.2 Symbolic reasoning for entity alignment

Given an aligned pair $(e_j, e_j')$, a new aligned pair $(e_i, e_i')$ can be inferred with confidence score $w_{p,p'}$ if they are each connected to the existing pair via a relational path $p$ and $p'$ respectively, formally:

$$w_{p,p'} : (e_j \equiv e_j') \wedge p(e_i, e_j) \wedge p'(e_i', e_j') \Longrightarrow (e_i \equiv e_i'), \tag{1}$$

where $p = |\mathcal{R}|^L, p' = |\mathcal{R}'|^L$ are a pair of paths each consisting of $L$ connected relations, and $w_{p,p'}$ measures the rule quality that considers the intra-KG structure and inter-KG structure, such as the indicative of each path, and the similarity between two paths. By instantiating such *rule* with the constants (real entities and relations) in the KG pair, a symbolic model predicts the label distribution of an entity pair $(e, e')$ by:

$$p_w(\boldsymbol{v}_{(e,e')}|\mathcal{G}, \mathcal{G}'), \quad \text{for } (e, e') \in \{\mathcal{O} \cup \mathcal{H}\}. \tag{2}$$

Using logic rules to infer the alignment probability can leverage the high-order structural information for effective alignment as well as provide interpretability. However, exact inference is intractable due to the massive amount of possible instantiated rules (exponential to $L$), limiting its applicability to real-world KGs.

## 3 Neuro-symbolic reasoning framework for entity alignment

### 3.1 Variational EM

Given a set of observed labels $\boldsymbol{v}_O$, our goal is to maximize the log-likelihood of these labels, i.e., $\log p_w(\boldsymbol{v}_O)$. Directly optimizing this objective is intractable because it requires computing an

integral over all the hidden variables. Instead, we optimize the evidence lower bound (ELBO) of the log-likelihood as follows:

$$p_w(\boldsymbol{v}_O) \geq E_{q(\boldsymbol{v}_H)}[\log p_w(\boldsymbol{v}_O, \boldsymbol{v}_H) - \log q(\boldsymbol{v}_H)] = \text{ELBO}(q, \boldsymbol{v}_O; w), \qquad (3)$$

here, $q(\boldsymbol{v}_H)$ is a variational distribution of the hidden variables $\boldsymbol{v}_H$. This inequality holds for all $q$ because $p_w(\boldsymbol{v}_O) = \text{ELBO}(q, \boldsymbol{v}_O; w) + D_{KL}(q(\boldsymbol{v}_H) \| p_w(\boldsymbol{v}_H \mid \boldsymbol{v}_O))$, where $D_{KL}(q(\boldsymbol{v}_H) \| p_w(\boldsymbol{v}_H \mid \boldsymbol{v}_O)) \geq 0$ is the KL-divergence between $q(\boldsymbol{v}_H)$ and $p_w(\boldsymbol{v}_H \mid \boldsymbol{v}_O)$. Under this framework, the log-likelihood $p_w(\boldsymbol{v}_O)$ can be optimized using an EM algorithm, an efficient method to find the maximum likelihood where the model depends on unobserved hidden variables: during the E-step, we fix $w$ and update the variational distribution $q$; during the M-step, we update $w$ to maximize the log-likelihood of all the entity pairs, i.e., $E_{q(\boldsymbol{v}_H)}[\log p_w(\boldsymbol{v}_O, \boldsymbol{v}_H)]$, as illustrated in Figure 1.

Explicitly representing the variational distribution $q$ is parameter intensive, which requires $\approx |\mathcal{E}||\mathcal{E}'|$ variables because the observed pairs are very sparse. To this end, we parameterize $q$ with a neural model as $q_\theta$, with $\theta$ being the parameters of the neural model.

### 3.2 E-step: inference

In this step, we fix $w$ and update $q_\theta$ to minimize the KL divergence $D_{KL}$. Directly minimizing the KL divergence is intractable, as it involves computing the entropy of $q_\theta$. Therefore, we follow [26] and optimize the reverse KL divergence of $q_\theta$ and $p_w$, leading to the following objective:

$$\phi_{\boldsymbol{v}_H, \theta} = \sum_{(e,e') \in \mathcal{H}} \mathbf{E}_{p_w(\boldsymbol{v}_{(e,e')}|\boldsymbol{v}_O)} q_\theta(\boldsymbol{v}_H). \qquad (4)$$

To optimize this objective, we first use the symbolic model with weighted rules to predict $p_w(\boldsymbol{v}_{(e,e')} \mid \boldsymbol{v}_O)$ for each $(e, e') \in \mathcal{H}$. If $p_w(\boldsymbol{v}_{(e,e')} \mid \boldsymbol{v}_O) > \delta$, where $\delta$ is a threshold, we treat this entity pair as a positive label; otherwise, we regard the pair as a negative pair that can be selected during negative sampling process of the neural model.

The observed labels can also be used as training data for supervised optimization. The objective is:

$$\phi_{\boldsymbol{v}_O, \theta} = \sum_{(e,e') \in \mathcal{O}} \log q_\theta(\boldsymbol{v}_{(e,e')} = 1). \qquad (5)$$

The final objective for $q_\theta$ is obtained by combining these two objectives: $\phi_\theta = \phi_{\boldsymbol{v}_H, \theta} + \phi_{\boldsymbol{v}_O, \theta}$.

### 3.3 M-step: rule weight update

In this step, we fix $q_\theta$ and update the rule weight $w$ to maximize $\text{ELBO}(q, \boldsymbol{v}_O; w)$. Since the right term of the ELBO in Equation 3 is constant when $q_\theta$ is fixed, the objective is equivalent to maximizing the left term $E_{q_\theta(\boldsymbol{v}_H)}[\log p_w(\boldsymbol{v}_O, \boldsymbol{v}_H)]$, which is the log-likelihood function.

Specifically, we start by predicting the labels of hidden variables using the current neural model. For each $(e, e') \in \mathcal{H}$, we predict the labels $\hat{\boldsymbol{v}}_{(e,e')}(\theta)$ and obtain the prediction set $\hat{\boldsymbol{v}}_H(\theta) = \{\hat{\boldsymbol{v}}_{(e,e')}(\theta)\}_{(e,e') \in \mathcal{H}}$. In this way, maximizing the likelihood practically becomes maximizing the following objective:

$$\phi_w = \log p_w(\boldsymbol{v}_O, \hat{\boldsymbol{v}}_H(\theta)). \qquad (6)$$

To obtain the pseudo-label $\hat{\boldsymbol{v}}_{(e,e')}$ using $q_\theta$, we employ the trained neural model to compute the matching score of any entity pair $(e, e') \in \mathcal{H}$. However, this strategy can easily introduce false positives into the pseudo-label set especially when the number of entities is large. To mitigate this, we consider one-to-one matching to sift only the most confident pairs. Practically, we first sort all pairs by their confidence score, then we annotate the pairs as positive following the order of the confidence. If a pair contains an entity observed in the annotated pairs, then this pair is skipped. This simple greedy strategy significantly reduces the amount of false positives.

## 4 Optimization and inference

### 4.1 Efficient optimization via logical deduction

Inference and learning with logic rules of length $L$ can be intractable, as the search space for paths grows exponentially with increasing $L$. To enhance reasoning efficiency, we decompose a rule in

Equation 1 using logic deduction, inspired by [28] in KG completion:

$$w_{p,p'}: \quad (e_j \equiv e'_j) \wedge \left( \bigwedge_{k=1}^{L} r_k(e_{k-1}, e_k) \right) \wedge \left( \bigwedge_{k=1}^{L} r'_k(e'_{k-1}, e'_k) \right) \implies (e_i \equiv e'_i). \quad (7)$$

Here $\bigwedge_{k=1}^{L} r_k(e_{k-1}, e_k)$ represents the path formed by $r_1, r_2, ..., r_L$ connecting $e_i$ to $e_j$ with $e_0 = e_i$ and $e_k = e_j$. This long rule can be reorganized as *the combination of a series of unit-length logic reasoning*:

$$w_{p,p'}: \quad (e_j \equiv e'_j) \wedge \left( \bigwedge_{k=1}^{L} \left[ r_k(e_{k-1}, e_k) \wedge r'_k(e'_{k-1}, e'_k) \right] \right) \implies (e_i \equiv e'_i). \quad (8)$$

In this way, each logic rule of length $L$ can be viewed as *a deductive combination of L short rules of length 1*. At each step, following [12], we perform one-step inference to update $p_w(\boldsymbol{v}_{(e,e')})$ for each $(e, e') \in \mathcal{H}$ by aggregating the alignment probability from neighbors:

$$1 - \prod_{\substack{(e,r,e_t)\in\mathcal{T}, \\ (e',r',e'_t)\in\mathcal{T}'}} \left(1 - \eta(r)p_{sub}(r \subseteq r')p_w(\boldsymbol{v}_{(e_t,e'_t)}))\right) \times \left(1 - \eta(r')p_{sub}(r' \subseteq r)p_w(\boldsymbol{v}_{(e_t,e'_t)}))\right), \quad (9)$$

where $\eta(r)$ is a relation pattern of $r$ measuring the uniqueness of $e$ through relation $r$ given a specified tail entity $e_t$, quantified by $\eta(r) = \frac{|\{e_t|(e_h,r,e_t)\in\mathcal{T}\}|}{|\{(e_h,e_t)|(e_h,r,e_t)\in\mathcal{T})\}|}$. $p_{sub}(r \subseteq r')$ denotes the probability that relation $r$ is a subrelation of $r'$. This technique enables inference with confidence by explicitly quantifying confidence $w$ during each inference step by introducing $\eta$ and $p_{sub}(r \subseteq r')$. Moreover, in this way, the update of the weight $w$ simplifies to updating $p_{sub}(r \subseteq r')$ during the M-step (Equation 6), as $\eta(r)$ for each relation $r$ is constant. In practice, the update of $p_{sub}(r \subseteq r')$ can be computed by:

$$\frac{\sum \left(1 - \prod_{(e'_h,r',e'_t)\in\mathcal{T}'} \left(1 - \boldsymbol{v}_{(e_h,e'_h)}\boldsymbol{v}_{(e_t,e'_t)}\right)\right)}{\sum \left(1 - \prod_{e'_h,e'_t\in\mathcal{E}'} \left(1 - \boldsymbol{v}_{(e_h,e'_h)}\boldsymbol{v}_{(e_t,e'_t)}\right)\right)}, \quad (10)$$

where $\boldsymbol{v}_{(e_h,e'_h)}$ and $\boldsymbol{v}_{(e_t,e'_t)}$ are labels (or pseudo-labels) from $\boldsymbol{v}_O \cup \hat{\boldsymbol{v}}_H(\theta)$.

After optimization, rule weights can be computed by taking the product of the importance scores $\eta$ of relations and the sub-relation probabilities of the corresponding relation pair:

$$w_{p,p'} := \prod_{k=1}^{L} \eta(r_k) \cdot \eta(r'_k) \cdot \frac{p_{sub}(r_k \subseteq r'_k) + p_{sub}(r'_k \subseteq r_k)}{2}. \quad (11)$$

In Appendix A.2, we provided a detailed complexity analysis, demonstrating that parameter complexity scales linearly with dataset size, while computational complexity is quadratic. Our implementation enables efficient execution through parallel computing and batch processing.

## 4.2 Inference with interpretability

To predict new alignments, there are two approaches: using the symbolic model or the neural model. The symbolic model infers alignment probabilities with the optimized weights $w$. Due to scalability concerns, symbolic methods generally adopt a lazy inference strategy that only preserves the confidently inferred pairs during inference. On the other hand, the neural model computes similarity scores for all entity pairs $(e, e') \in \mathcal{H}$ using the learned parameters $\theta$, generating a ranked candidate list for each entity.

The evaluation of these models thus differs. Symbolic models are generally evaluated by precision, recall, and F1-score for their binary outputs, while neural models are assessed using hit@k and mean reciprocal ranks (MRR) for their ranked candidate lists. Following the practices in [14] and [24], we unify the evaluation metrics by treating the recall metric of symbolic models as equivalent to hit@1, facilitating comparison with neural models.

To enhance the interpretability of predictions, we adapt the optimized symbolic model into an explainer. For any given entity pair, the explainer generates a set of supporting rule path pairs that

justify their alignment, each associated with a confidence score indicating its significance. The explainer operates in two modes: ❶ hard-anchor mode, which generates supporting paths only from prealigned pairs, and ❷ soft-anchor mode, which includes paths from both prealigned and inferred pairs, providing more informative interpretations.

By integrating a breadth-first search algorithm (detailed in Appendix A.3), the explainer efficiently generates high-quality interpretations. For truly aligned pairs, it typically produces high-confidence interpretations, while for non-aligned pairs, the interpretations may result in an empty set (indicating no supporting evidence) or have low confidence scores. See Figure 4 for a visualized comparison.

## 5 Experiments

### 5.1 Experimental settings

**Datasets.** Main experiments use the DBP15K dataset, comprising three cross-lingual KG pairs: JA-EN, FR-EN, and ZH-EN. The original full version [29] of DBP15K resembles real-world KGs, posing challenges for GCN-based models due to sparsity and scale. Recent GCN-based models [19, 16, 17, 24] remove low-degree entities to get a smaller version with higher average degree. For thorough evaluation, we utilize **both full and condensed versions**. Dataset statistics are provided in Appendix B.1. For additional experiments on large KGs, we employ OpenEA [30] and DBP1M.

Two different dataset split strategies are used in the EA literature: ❶ a 3:7 train/test split, and ❷ a 5-fold cross-validation scheme with a 2:1:7 ratio for training, validation, and test sets, as used in OpenEA [30]. We adopt the latter for all algorithms to ensure fair comparison.

**Baselines and metrics.** Baseline models include seven neural models – GCNAlign [19], AlignE, BootEA [16], RREA [17], Dual-AMN [18], LightEA [20], PEEA [31], one symbolic models – PARIS [12], and two neuro-symbolic models – PRASE [14], EMEA [24]. We use Hit@1, Hit@10, and MRR as the evaluation metrics. For PARIS and PRASE that have binary outputs, we report their recall as Hit@1, following [24]. For RREA, Dual-AMN, and LightEA, which offer both basic and iterative versions, we adopt the iterative ones due to their generally superior performance.

**Hyperparameters.** NeuSymEA involves two main hyperparameters: the number of EM iterations and the symbolic model's threshold $\delta$ for selecting positive pairs. We tune them on the validation set, searching $\delta$ in $0.6, 0.7, 0.8, 0.9, 0.95, 0.98, 0.99$ and the number of iterations from 1 to 9. As shown in Section 5.3, NeuSymEA is robust to $\delta$ and converges quickly.

### 5.2 Results

#### 5.2.1 Comparison with baselines

Table 1 compares NeuSymEA with baseline models on two versions of the DBP15K dataset: the full and the condensed version. Results for PRASE and EMEA on the condensed DBP15K are sourced from the original EMEA paper. The results yield three key observations:

**First, NeuSymEA surpasses both symbolic and neural models.** By integrating symbolic reasoning with neural representations, it: 1) captures multi-hop relational structures across KGs using rules; and 2) learns effective entity representations to compute pairwise similarities. **Second, NeuSymEA outperforms other neuro-symbolic models.** This improvement can be largely attributed to the model objective design in our framework. While PRASE and EMEA treat the symbolic and neural models as separate components, NeuSymEA unifies them under a joint probability objective. This enables joint optimization via Variational EM, yielding a more coherent and convergent solution with superior performance. **Finally, NeuSymEA demonstrates robustness across both full and condensed datasets.** Comparisons between two groups of results offer an interesting insight: neural models experience significant performance degradation in the full version of DBP15K (e.g., MRR of Dual-AMN decreases from 0.815 to 0.717 on JA-EN), while symbolic models, in contrast, show improvements. We attribute this to their different matching mechanisms: (1) neural models rely on *entity-level matching*, which is sensitive to dataset size. The full DBP15K includes more low-degree entities, increasing similar embeddings and reducing precision. (2) Symbolic models use *path-level matching*, which is less affected by dataset size but vulnerable to substructure heterogeneity. The full DBP15K's additional connections via long-tail entities enhance rule-mining, boosting symbolic

Table 1: Entity alignment results on DBP15K dataset. The suffixes "-D" and "-L" indicate the use of Dual-AMN and LightEA as the neural models. The results of RREA and EMEA are omitted on the full dataset due to an OOM (Out of Memory) error.

| Category | Model | JA-EN | | | FR-EN | | | ZH-EN | | |
|---|---|---|---|---|---|---|---|---|---|---|
| | | Hit@1 | Hit@10 | MRR | Hit@1 | Hit@10 | MRR | Hit@1 | Hit@10 | MRR |
| *Results on the full DBP15K dataset* | | | | | | | | | | |
| **Neural** | GCNAlign | 0.221 | 0.461 | 0.302 | 0.205 | 0.475 | 0.295 | 0.189 | 0.438 | 0.271 |
| | BootEA | 0.454 | 0.782 | 0.564 | 0.443 | 0.799 | 0.564 | 0.486 | 0.814 | 0.600 |
| | AlignE | 0.356 | 0.715 | 0.476 | 0.346 | 0.731 | 0.475 | 0.333 | 0.690 | 0.453 |
| | Dual-AMN | 0.627 | 0.883 | 0.717 | 0.652 | 0.908 | 0.744 | 0.650 | 0.884 | 0.732 |
| | LightEA | 0.736 | 0.894 | 0.793 | 0.782 | 0.919 | 0.832 | 0.725 | 0.874 | 0.779 |
| **Symbolic** | PARIS | 0.589 | - | - | 0.618 | - | - | 0.603 | - | - |
| **Neuro-symbolic** | PRASE | 0.611 | - | - | 0.647 | - | - | 0.652 | - | - |
| **Ours** | NeuSymEA-D | **0.806** | **0.942** | **0.855** | 0.827 | **0.952** | **0.871** | **0.801** | **0.925** | **0.843** |
| | NeuSymEA-L | 0.781 | 0.907 | 0.826 | **0.834** | 0.937 | **0.871** | 0.785 | 0.894 | 0.825 |
| *Results on the condensed DBP15K dataset* | | | | | | | | | | |
| **Neural** | GCNAlign | 0.331 | 0.662 | 0.443 | 0.325 | 0.688 | 0.446 | 0.335 | 0.653 | 0.442 |
| | BootEA | 0.530 | 0.829 | 0.631 | 0.579 | 0.872 | 0.961 | 0.575 | 0.847 | 0.668 |
| | AlignE | 0.433 | 0.783 | 0.552 | 0.457 | 0.821 | 0.580 | 0.474 | 0.806 | 0.587 |
| | RREA | 0.749 | **0.935** | 0.818 | 0.797 | **0.958** | 0.859 | 0.762 | **0.938** | 0.827 |
| | Dual-AMN | 0.750 | 0.927 | 0.815 | 0.793 | 0.954 | 0.854 | 0.756 | 0.919 | 0.816 |
| | LightEA | 0.778 | 0.911 | 0.828 | 0.827 | 0.943 | 0.830 | 0.770 | 0.894 | 0.816 |
| | PEEA | 0.703 | 0.912 | 0.777 | 0.748 | 0.937 | 0.815 | 0.726 | 0.905 | 0.790 |
| **Symbolic** | PARIS | 0.565 | - | - | 0.584 | - | - | 0.543 | - | - |
| **Neuro-symbolic** | PRASE | 0.580 | - | - | 0.622 | - | - | 0.593 | - | - |
| | EMEA | 0.736 | - | 0.807 | 0.773 | - | 0.841 | 0.748 | - | 0.815 |
| **Ours** | NeuSymEA-D | 0.805 | **0.930** | 0.849 | 0.835 | 0.953 | 0.879 | **0.815** | **0.926** | **0.855** |
| | NeuSymEA-L | **0.811** | 0.928 | **0.854** | **0.858** | 0.954 | **0.894** | 0.804 | 0.904 | 0.840 |

model performance. By integrating symbolic reasoning with KG embeddings, NeuSymEA overcomes these limitations, ensuring robustness to variations in dataset scale and structure.

### 5.2.2 Evolution of rules and embeddings

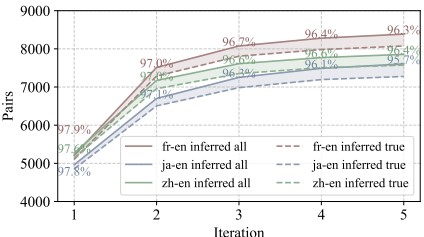 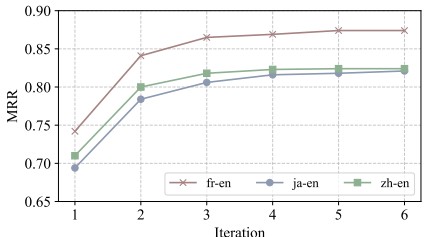

Figure 2: (Left) Evolution of rule inferred pairs, with solid lines representing total inferred pairs and dashed lines representing true inferred. The shaded areas indicate the number of false pairs. Precision values are annotated at each data point. (Right) Convergence of MRR of the neural model.

We study how rules and embeddings evolve and interact with each other during the EM steps, with results shown in Figure 2. Results in the left subplot indicate that in each EM iteration, the number of rule-inferred pairs grows consistently with high precision, implying that the embedding model continuously improves the inference performance of rules. These precise pairs, in turn, enhance the performance of the neural model. As shown in the right subfigure, the MRR of the neural model converges within a few iterations.

### 5.2.3 Scalability on large datasets

Figure 3 illustrate the scalability of NeuSymEA. We evaluated its hit@1 accuracy and runtime efficiency across datasets of varying entity sizes, namely DBP15K, OpenEA100K, and DBP1M.

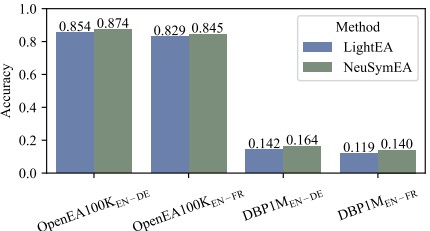 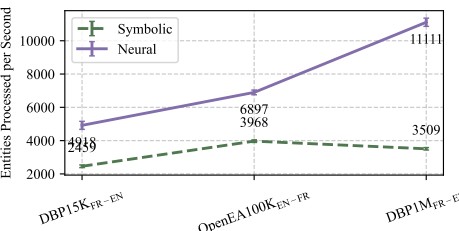

Figure 3: Scalability analysis on large scale KGs. (Left) Hit@1 alignment performance on large KGs. (Right) Per-second processed entities of neural and symbolic components on different scales of KGs.

**Performance scalability.** In the left subfigure, we compare NeuSymEA with LightEA - the only strong baseline capable of processing large-scale knowledge graphs with million-scale entities. The results demonstrate NeuSymEA's superior scalability in terms of performance.

**Runtime Scalability.** We separately evaluate the runtime performance of NeuSymEA's neural and symbolic reasoning components. For the neural component, efficiency (entities processed per second) increases with dataset size. We attribute this to higher GPU utilization on larger datasets, which enhances computational efficiency. Conversely, the symbolic component's efficiency initially increases but then slightly decreases. Upon investigation, we found that this phenomenon is related to the multiprocessing and batch-processing implementation. For smaller datasets (e.g., DBP15K), the overhead from process initialization and termination is significant. As dataset size grows to OpenEA100K, this overhead becomes negligible relative to inference runtime, leading to an efficiency improvement. When moving from OpenEA100K to DBP1M, the per-second processed entities slightly decline as the quadratic complexity of inference computation dominates.

### 5.2.4 Interpretations by the explainer

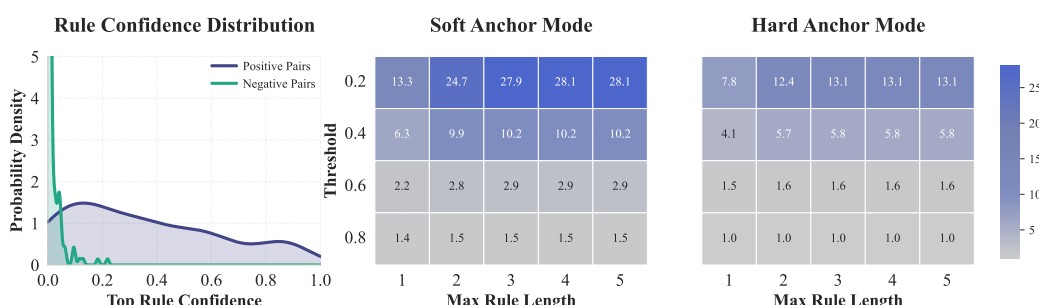

Figure 4: (Left) Probability density of the top supporting rule's confidence; (Middle) Number of supporting rules relative to the maximum rule lengths under the soft anchor mode; (Right) Number of supporting rules relative to the maximum rule lengths under the hard anchor mode.

We investigate the interpretations generated by the explainer on the FR-EN dataset in Figure 4. The left subfigure displays the probability density of confidence scores for supporting rules associated with entity pairs. Positive pairs come from the test set, while negative pairs are generated by randomly replacing one entity in each positive pair. The distinct confidence distributions show that positive pairs generally have stronger alignment evidence, as expected. Upon further examination, we found that many test pairs are isolated, i.e., they lack directly aligned neighbors. Despite this, NeuSymEA successfully generates supporting rules for isolated pairs by exploiting multi-hop dependencies.

The middle and right subfigures examine the impact of rule length on the explainer's effectiveness, showing the number of supporting rules for positive pairs as the maximum rule length increases. In soft anchor mode, the explainer produces more high-quality supporting rules than in hard anchor mode by using inferred pairs as complementary anchors, mitigating substructure sparsity. While longer maximum rule lengths yield more high-quality supporting rules, however, the long rules have a lower confidence distribution compared to short rules. This can be attributed to our method for calculating confidence: the logical deduction-based approach computes a rule's confidence as the

product of the confidences of its decomposed unit-length sub rules (as in Equation 11). For instance, a rule comprising two unit-length sub-rules, each with a confidence of 0.8, has a combined confidence of $0.8 \times 0.8 = 0.64$. Thus, confidence scores typically decrease as rule length increases.

Table 2: Examples of supporting rules for query pairs in FR-EN. Anchor pairs are shown in bold.

| Query Pair | Supporting Rule | Confidence |
|---|---|---|
| Maison_de_Savoie
House_of_Savoy | (Humbert_II_(roi_d'Italie), dynastie, Maison_de_Savoie), (Humbert_II_(roi_d'Italie), conjoint, **Marie-José_de_Belgique**)
(Umberto_II_of_Italy, house, House_of_Savoy), (Umberto_II_of_Italy, spouse, **Marie_José_of_Belgium**) | 0.80 |
| Légion_espagnole
Spanish_Legion | (Légion_espagnole, commandantHistorique, Francisco_Franco), (Francisco_Franco, conjoint, **Carmen_Polo**)
(Spanish_Legion, notableCommanders, Francisco_Franco), (Francisco_Franco, spouse, **Carmen_Polo,_1st_Lady_of_Meirás**) | 0.59 |
| Premier_ministre_du_Danemark
Prime_Minister_of_Denmark | (Premier_ministre_du_Danemark, titulaireActuel, **Lars_Løkke_Rasmussen**)
(Prime_Minister_of_Denmark, incumbent, **Lars_Løkke_Rasmussen**) | 0.79 |

### 5.2.5 Robustness in low resource scenario

Table 3: Comprehensive results with different ratios of training data, with best shown in bold.

| Dataset | Model | 1% | | | 5% | | | 10% | | | 20% | | |
|---|---|---|---|---|---|---|---|---|---|---|---|---|---|
| | | H@1 | H@10 | MRR | H@1 | H@10 | MRR | H@1 | H@10 | MRR | H@1 | H@10 | MRR |
| JA-EN | AlignE | 0.007 | 0.034 | 0.016 | 0.080 | 0.268 | 0.143 | 0.244 | 0.588 | 0.356 | 0.433 | 0.783 | 0.552 |
| | BootEA | 0.010 | 0.040 | 0.021 | 0.379 | 0.683 | 0.481 | 0.468 | 0.779 | 0.573 | 0.530 | 0.829 | 0.631 |
| | GCNAlign | 0.029 | 0.128 | 0.063 | 0.127 | 0.368 | 0.206 | 0.209 | 0.515 | 0.310 | 0.331 | 0.662 | 0.443 |
| | PARIS | 0.145 | - | - | 0.340 | - | - | 0.450 | - | - | 0.565 | - | - |
| | PRASE | 0.163 | - | - | 0.432 | - | - | 0.508 | - | - | 0.580 | - | - |
| | Dual-AMN | 0.239 | 0.519 | 0.334 | 0.509 | 0.795 | 0.611 | 0.652 | 0.887 | 0.738 | 0.750 | 0.927 | 0.815 |
| | RREA | 0.253 | 0.486 | 0.332 | 0.558 | 0.830 | 0.653 | 0.672 | **0.903** | 0.756 | 0.789 | **0.956** | 0.853 |
| | LightEA | 0.291 | 0.514 | 0.363 | 0.627 | 0.806 | 0.689 | 0.714 | 0.874 | 0.771 | 0.778 | 0.911 | 0.828 |
| | EMEA | 0.411 | - | 0.488 | 0.630 | - | 0.710 | 0.688 | - | 0.764 | 0.736 | - | 0.807 |
| | PEEA | 0.242 | 0.519 | 0.333 | 0.490 | 0.785 | 0.589 | 0.612 | 0.834 | 0.679 | 0.703 | 0.912 | 0.777 |
| | NeuSymEA-D | 0.481 | 0.684 | 0.550 | 0.692 | 0.855 | 0.749 | 0.742 | 0.895 | 0.796 | 0.835 | 0.953 | 0.879 |
| | NeuSymEA-L | **0.632** | **0.779** | **0.683** | **0.733** | **0.870** | **0.781** | **0.773** | 0.900 | **0.818** | **0.858** | 0.954 | **0.894** |
| FR-EN | AlignE | 0.008 | 0.040 | 0.019 | 0.127 | 0.408 | 0.217 | 0.347 | 0.733 | 0.475 | 0.457 | 0.821 | 0.580 |
| | BootEA | 0.009 | 0.041 | 0.020 | 0.418 | 0.746 | 0.529 | 0.490 | 0.809 | 0.598 | 0.579 | 0.872 | 0.681 |
| | GCNAlign | 0.027 | 0.119 | 0.058 | 0.133 | 0.388 | 0.215 | 0.215 | 0.539 | 0.321 | 0.325 | 0.688 | 0.446 |
| | PARIS | 0.195 | - | - | 0.401 | - | - | 0.479 | - | - | 0.584 | - | - |
| | PRASE | 0.227 | - | - | 0.514 | - | - | 0.575 | - | - | 0.633 | - | - |
| | Dual-AMN | 0.293 | 0.631 | 0.407 | 0.598 | 0.886 | 0.703 | 0.717 | 0.928 | 0.797 | 0.793 | 0.954 | 0.854 |
| | RREA | 0.289 | 0.583 | 0.389 | 0.628 | 0.895 | 0.725 | 0.717 | 0.932 | 0.796 | 0.789 | **0.956** | 0.853 |
| | LightEA | 0.430 | 0.663 | 0.509 | 0.723 | 0.885 | 0.781 | 0.779 | 0.914 | 0.828 | 0.827 | 0.943 | 0.870 |
| | EMEA | 0.480 | - | 0.565 | 0.677 | - | 0.757 | 0.727 | - | 0.802 | 0.773 | - | 0.841 |
| | PEEA | 0.285 | 0.588 | 0.385 | 0.552 | 0.812 | 0.642 | 0.665 | 0.875 | 0.738 | 0.748 | 0.937 | 0.815 |
| | NeuSymEA-D | 0.642 | 0.833 | 0.709 | 0.768 | 0.916 | 0.820 | 0.811 | **0.939** | 0.856 | 0.835 | 0.953 | 0.879 |
| | NeuSymEA-L | **0.737** | **0.874** | **0.785** | **0.806** | **0.921** | **0.848** | **0.827** | 0.937 | **0.867** | **0.858** | 0.954 | **0.894** |
| ZH-EN | AlignE | 0.006 | 0.033 | 0.016 | 0.127 | 0.368 | 0.206 | 0.296 | 0.635 | 0.407 | 0.474 | 0.806 | 0.587 |
| | BootEA | 0.006 | 0.029 | 0.014 | 0.396 | 0.689 | 0.495 | 0.498 | 0.782 | 0.594 | 0.575 | 0.847 | 0.668 |
| | GCNAlign | 0.041 | 0.155 | 0.080 | 0.147 | 0.396 | 0.229 | 0.225 | 0.519 | 0.323 | 0.335 | 0.653 | 0.442 |
| | PARIS | 0.059 | - | - | 0.333 | - | - | 0.429 | - | - | 0.543 | - | - |
| | PRASE | 0.241 | - | - | 0.461 | - | - | 0.522 | - | - | 0.593 | - | - |
| | Dual-AMN | 0.375 | 0.666 | 0.480 | 0.582 | 0.830 | 0.672 | 0.676 | 0.892 | 0.755 | 0.756 | 0.918 | 0.816 |
| | RREA | 0.316 | 0.564 | 0.403 | 0.605 | **0.858** | 0.696 | 0.686 | **0.901** | 0.765 | 0.760 | **0.934** | 0.823 |
| | LightEA | 0.507 | 0.673 | 0.565 | 0.670 | 0.819 | 0.723 | 0.727 | 0.860 | 0.775 | 0.770 | 0.894 | 0.816 |
| | EMEA | 0.517 | - | 0.591 | 0.665 | - | 0.738 | 0.706 | - | 0.777 | 0.748 | - | 0.815 |
| | PEEA | 0.288 | 0.586 | 0.388 | 0.532 | 0.801 | 0.622 | 0.649 | 0.871 | 0.710 | 0.725 | 0.905 | 0.790 |
| | NeuSymEA-D | 0.589 | 0.750 | 0.645 | 0.704 | 0.856 | 0.757 | 0.763 | 0.897 | 0.809 | **0.815** | 0.926 | **0.855** |
| | NeuSymEA-L | **0.676** | **0.799** | **0.720** | **0.735** | 0.858 | **0.779** | **0.773** | 0.882 | **0.811** | 0.804 | **0.934** | 0.841 |

Table 3 demonstrates the model performance under low-resource settings. As the percentage of training data decreases, all models experience noticeable drops in Hit@1 performance. Despite this, NeuSymEA exhibits remarkable robustness across all datasets, consistently outperforming other models. Notably, with only 1% of pairs used as training data, NeuSymEA achieves a Hit@1 score exceeding 0.7 on FR-EN, rivaling or even surpassing the performance of some state-of-the-art models trained on 20% of the data.

### 5.3 Parameter analysis

We present the hit@1 performance of NeuSymEA across three datasets, varying hyperparameters, illustrated by a three-dimensional graph. The threshold hyperparameter $\delta$ is explored within the set {0.6, 0.7, 0.8, 0.9, 0.95, 0.98, 0.99}, while the number of EM iterations ranges from 1 to 9. Performance levels are indicated using a colormap. Performance sensitivity analysis in Figure 5

reveals that for all datasets, performance generally improves as the iteration increases. On the other hand, the performance is less sensitive to the threshold $\delta$.

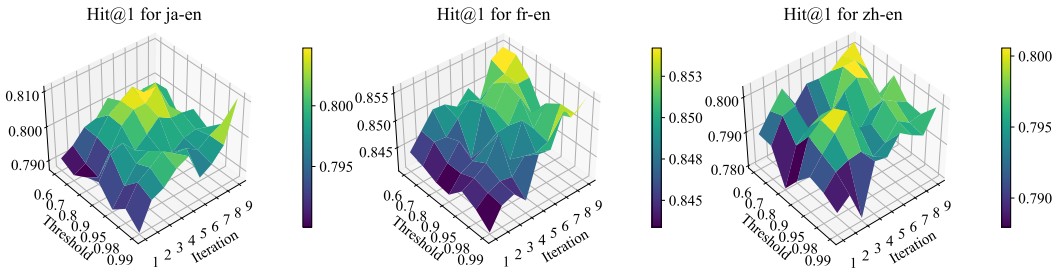

Figure 5: Performance sensitivity to hyperparameters iteration and threshold $\delta$.

# 6 Related work

**Neuro-symbolic reasoning on knowledge graphs.** Neuro-symbolic methods aim to combine symbolic reasoning with neural representation learning, leveraging the precision and interpretability of symbolic approaches alongside the scalability and high recall of neural methods. In KG completion task, [32] and [33] employ horn rules to regularize the learning of KG embeddings; [34] and [28] model the rule-based predictions as distributions conditioned on the input relational sequences, and parameterize these distributions using a recurrent neural network; [26], [27] and [35] models the joint probability of the neural model and the symbolic model with a Markov random field, and employ gradient descent for weight updates. Despite extensive advancements of neuro-symbolic reasoning in KG completion, these studies only consider single-KG structures, thus cannot be directly adopted to entity alignment which requires consideration of inter-KG structures.

**Entity alignment.** Recent models have sought to combine symbolic and neural approaches for entity alignment. For instance, [14] enhances probabilistic reasoning with KG embeddings to measure entity-level and relation-level similarities. [24] implements self-bootstrapping with pseudo-labeling in a neural framework, using rules to choose confident pseudo-labels. However, it relies solely on unit-length rules, which restricts its effectiveness for long-tail entities. In contrast, we employ logic deduction to scale symbolic reasoning with long rules of any length. Unlike existing work that separately handles two reasoning components, our work models the unified joint probability of symbolic and neural inference within a markov random field. The symbolic component captures intra-KG and inter-KG structures using weighted logic rules, while the neural model learn expressive patterns in the embedding space.

# 7 Limitations

NeuSymEA is currently designed for EA between two KGs. Extending it to align multiple KGs simultaneously may require iterative pairwise alignments, which could be inefficient. A more sophisticated optimization paradigm is needed to adapt NeuSymEA for scalable multi-KG alignment.

# 8 Conclusions

We presented NeuSymEA, a unified and extensible neuro-symbolic framework for entity alignment. By unifying neural and symbolic reasoning, NeuSymEA addresses the challenges of substructure heterogeneity, sparsity, and uncertainty in real-world KGs. Empirical results demonstrate NeuSymEA's clear improvements over baselines and robustness under limited resources. By delivering interpretable alignment predictions with uncertainty scores, NeuSymEA advances knowledge fusion, enabling effective and trustworthy entity alignment.

## Acknowledgement

The work described in this paper was fully supported by a grant from the Research Grants Council of the Hong Kong Special Administrative Region, China (Project No. PolyU 25208322).

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

# A  Notations and algorithms

## A.1  Notations

Table 4: Notations

| Notation | Description |
|---|---|
| $\mathcal{G}, \mathcal{G}'$ | The source and target knowledge graphs, respectively |
| $\mathcal{E}, \mathcal{E}'$ | The sets of entities in $\mathcal{G}$ and $\mathcal{G}'$, respectively |
| $\mathcal{R}, \mathcal{R}'$ | The sets of relations in $\mathcal{G}$ and $\mathcal{G}'$, respectively |
| $\mathcal{T}, \mathcal{T}'$ | The sets of relational triplets in $\mathcal{G}$ and $\mathcal{G}'$, respectively |
| $\mathcal{O}$ | The set of observed aligned entity pairs between two knowledge graphs $\mathcal{G}$ and $\mathcal{G}'$ |
| $\mathcal{H}$ | Set of unobserved entity pairs, i.e., $\mathcal{E} \times \mathcal{E}' \backslash \mathcal{O}$ |
| $\boldsymbol{v}_{(e,e')}$ | Binary indicator variable for an entity pair $(e, e')$, where $\boldsymbol{v}_{(e,e')} = 1$ indicates alignment |
| $w_{p,p'}$ | Confidence score of a rule-inferred alignment based on paths $p$ and $p'$ |
| $p_w(\boldsymbol{v}_{(e,e')}|\mathcal{G}, \mathcal{G}')$ | Probability distribution of the alignment indicator $\boldsymbol{v}_{(e,e')}$ given knowledge graphs $\mathcal{G}$ and $\mathcal{G}'$ |
| $\theta$ | Parameters of the neural model |
| $\delta$ | Threshold to select positive pair from the symbolic model |
| $\eta(r)$ | Relation pattern measuring the uniqueness of an entity through relation $r$ |

## A.2  Complexity analysis of the symbolic reasoning

In the following, we present the analysis of runtime complexity and parameter complexity one by one.

### A.2.1  Runtime complexity

In variational inference, the process of learning and inferring long rules (Equation 7) is simplified by decomposing them into unit-length rules (Equation 8). Consequently, rule weight learning (Equation 10) is only conducted for unit-length rules. The inference process for an $L$-length rule is then estimated by iteratively applying inference steps with unit-length rules (Equation 9) for $L$ iterations. This strategy effectively avoids the exponential search space associated with longer rules, making the computational complexity of the inference linear with respect to the rule length $L$.

Each iteration of reasoning with unit-length rules comprises an inference step (Equation 9) and a rule-weight learning step (Equation 10). These steps require computing the matching probability for all possible entity pairs and relation pairs, respectively. As a result, the computational complexity of the inference step and the weight updating step are $O(|\mathcal{E}||\mathcal{E}'|)$ and $O(|\mathcal{R}||\mathcal{R}'|)$, respectively.

Thus, the total **computational complexity** for reasoning with an $L$-length rule is $O(L \cdot (|\mathcal{E}||\mathcal{E}'| + |\mathcal{R}||\mathcal{R}'|))$. Given that entity sizes are typically much larger than relation sizes, this complexity can be approximated as $O(L \cdot |\mathcal{E}||\mathcal{E}'|)$.

Notably, the computations involved in Equation 9 and Equation 10 can be accelerated through parallel processing, which we have implemented. This optimization reduces the **runtime complexity** to $O\left(L \cdot \frac{|\mathcal{E}||\mathcal{E}'|}{n}\right)$, where $n$ represents the number of CPU cores available for parallelization.

### A.2.2  Parameter complexity

The total number of alignment probabilities for all entity pairs is $|\mathcal{E}||\mathcal{E}'|$, which is large when the entity sizes increase. We adopt a lazy inference strategy to enhance parameter efficiency. This strategy involves only saving the alignment probabilities of the most probable alignments:

$$\left\{ p_w(v_{(e_i,e_i')}), |, e_i \in \mathcal{E}, e_i' \in \mathcal{E}', p_w(v_{(e_i,e_i')}) = \max\left(\max_{e \in \mathcal{E}} p_w(v_{(e,e_i')}), \max_{e' \in \mathcal{E}'} p_w(v_{(e_i,e')})\right)\right\} \quad (12)$$

Probabilities of other entity pairs can be inferred from these saved alignment probabilities using Equation 9. In this way, **parameter complexity** is reduced to $O(\max(|\mathcal{E}| + |\mathcal{E}'|))$.

## A.3 Pseudo-code of Explainer

Below is the pseudo-code of how the explainer generates supporting rules as interpretations for the query pair. It consists of two stages: searching reachable anchor pairs, and parsing rule paths as well as calculating rule confidences.

---

**Algorithm 1** Generating Interpretations for the Queried Entity Pair with Weighted Rules

**Inputs:** Subrelation probabilities $p_{sub}(r \subseteq r'), p_{sub}(r' \subseteq r)$ for $r, r' \in \mathcal{R}$; Knowledge Graph pair $(\mathcal{G}, \mathcal{G}')$; Maximum rule length $\mathcal{L}$; Anchor pairs $\mathcal{A}$ with source-to-target mapping S2T and target-to-source mapping T2S; Query entity pair $(e_q, e_q')$

**Outputs:** Ranked rules based on confidence

**1. Search Reachable Anchor Pairs within Max Depth $\mathcal{L}$**

$RN \leftarrow \text{BFS}(e_q, \mathcal{G}, \mathcal{L})$  /* Search reachable neighbors of $e_q$ using breadth-first search, max depth $\mathcal{L}$ */

$RN' \leftarrow \text{BFS}(e_q', \mathcal{G}', \mathcal{L})$  /* Search reachable neighbors of $e_q'$ using breadth-first search, max depth $\mathcal{L}$ */

$RN_a \leftarrow RN \cup \text{T2S}(RN'; \mathcal{A})$  /* Find source nodes of reachable anchor pairs using hash mapping */

$RA \leftarrow \{(e, \text{S2T}(e; \mathcal{A})) \mid e \in RN_a\}$  /* Identify reachable anchor pairs */

**2. Parse and Rank Rules Based on Confidence**

**for** $\forall (e, e') \in RA$ **do**

 Extract paths: $p(e, e_q) = r_1 \wedge r_2 \wedge \ldots, p'(e', e_q') = r_1' \wedge r_2' \wedge \ldots$

 **if** $|p(e, e_q)| \neq |p'(e', e_q')|$ **then**

  $w_{p(e,e_q),p'(e',e_q')} \leftarrow 0$  /* If path lengths don't match, rule confidence is 0 */

 **else**

  $w_{p(e,e_q),p'(e',e_q')} \leftarrow \prod_{i=1}^{|p|} \eta(r_i) \cdot \eta(r_i') \cdot \frac{p_{sub}(r_i \subseteq r_i') + p_{sub}(r_i' \subseteq r_i)}{2}$  /* Compute rule confidence by

  products of subrelation probabilities and relation functionalities */

 **end if**

**end for**

Sort the rules $(p, p')$ by $w_{p,p'}$ in descending order

**Return** the ranked rules

---

# B Experimental details

## B.1 Dataset statistics

The DBP15K dataset, designed for cross-lingual knowledge graph alignment, has two versions: full and condensed. The original full version resembles real-world knowledge graphs, including comprehensive data across three language pairs with many sparsely connected low-degree entities. The condensed version, derived by JAPE and adopted by later methods (GCNAlign, RREA, Dual-AMN, LightEA), removes these low-degree entities and their connected triples to create a smaller (and higher average degree) dataset suitable for GCN-based methods. Detailed information about the two dataset versions can be found in the "dataset" section of readme in JAPE's official implementation[2].

The dataset statistics of them are shown in Table 5 and Table 6. In this paper, we adopt both versions of DBP15K for comprehensive evaluation. Specifically, the full dataset is sparser and larger in scale in scale due to the inclusion of low-degree entities, thus suitable for evaluating the models' robustness to sparsity and large scale.

Table 5: Data statistics of the full DBP15K dataset.

| Datasets | KG | Entities | Relations | Rel. Triplets | Aligned Entity Pairs |
|---|---|---|---|---|---|
| ZH-EN | Chinese (zh) | 66,469 | 2,830 | 153,929 | 15,000 |
|  | English (en) | 98,125 | 2,317 | 237,674 |  |
| JA-EN | Japanese (ja) | 65,744 | 2,043 | 164,373 | 15,000 |
|  | English (en) | 95,680 | 2,096 | 233,319 |  |
| FR-EN | French (fr) | 66,858 | 1,379 | 192,191 | 15,000 |
|  | English (en) | 105,889 | 2,209 | 278,590 |  |

---

[2] https://github.com/nju-websoft/JAPE

Table 6: Data statistics of the condensed DBP15K dataset.

| Datasets | KG | Entities | Relations | Rel. Triplets | Aligned Entity Pairs |
|---|---|---|---|---|---|
| ZH-EN | Chinese (zh) | 19,388 | 1,701 | 70,414 | 15,000 |
| | English (en) | 19,572 | 1,323 | 95,142 | |
| JA-EN | Japanese (ja) | 19,814 | 1,299 | 77,214 | 15,000 |
| | English (en) | 19,780 | 1,153 | 93,484 | |
| FR-EN | French (fr) | 19,661 | 903 | 105,998 | 15,000 |
| | English (en) | 19,993 | 1,208 | 115,722 | |

## B.2 Efficiency analysis

To provide a comprehensive understanding of the computational efficiency of our model, we report the runtime and memory usage during the experiments. The results, as summarized in Table Table 7, demonstrate that our model achieves efficient performance with a runtime of 15 minutes, a memory consumption of 868 MB, and a GPU memory usage of 4.33 GB. These metrics highlight the practicality of our approach in terms of resource utilization. Hardware configurations of the experiments are presented in Table 8.

Table 7: Runtime and Memory Usage

| Runtime | Memory | GPU Memory |
|---|---|---|
| 15 minutes | 868 MB | 4.33 GB |

Table 8: Machine configuration.

| Component | Specification |
|---|---|
| GPU | NVIDIA GeForce RTX 3090 |
| CPU | Intel(R) Xeon(R) Silver 4214R CPU @ 2.40GHz |

