# OpenReview forum: "NeuSymEA: Neuro-symbolic Entity Alignment via Variational Inference"
_NeurIPS.cc/2025/Conference — NeurIPS 2025 poster_

### Official Review · Reviewer_KxkU · 2025-06-13

**Clarity:** 2
**Significance:** 2
**Originality:** 3
**Rating:** 4
**Confidence:** 3

**Summary:**

This paper proposes a principled neuro-symbolic reasoning framework via variational EM. It provides efficient optimization through logical decomposition, allowing effective reasoning over large knowledge graphs. The paper also proposes interpretable inference with Hard-anchor and Soft-anchor modes and provides empirical validation demonstrating NeuSymEA’s high alignment accuracy and informative rule-based interpretations on benchmark datasets.

**Questions:**

1. LLMs contain large amount of world knowledge and thus may be easily applied to the EA task. Why didn't you consider this as a baseline or additional source of information in the framework?
2. eq. 1 and 2: It's unclear how the confidence and probability is inferred in the symbolic setting.
3. line 194: What exactly does the split mean? is it a split over entities or relations? how would it work if you apply something learned from one dataset to another?

**Ethical Concerns:**

["NO or VERY MINOR ethics concerns only"]

**Final Justification:**

The authors answered my questions and justified my rating.

**Limitations:**

yes

**Quality:**

3

**Strengths And Weaknesses:**

Quality: The proposed neuro-symbolic reasoning framework is derived in a principled way, although some assumptions and simplifications are applied. The experimental results seem to show the effectiveness of the proposed approach.

Clarity: Mostly clear but can benefit from additional information.

Significance: KG is no longer something critical in the LLM era. For this reason, the work is less important than before. In addition, If LLM is used, the performance of entity alignment may be further improved.

Originality: The proposed neuro-symbolic reasoning framework, although still falls into a standard framework, is sufficiently novel in the way the loss is selected and the optimization process is conducted.

---

> ### Author Rebuttal · Authors · 2025-07-31
>
> Thank you for your detailed review and the constructive questions. Below are our responses.
>
> ##  **[Q1]**
> ```
> LLMs contain large amount of world knowledge and thus may be easily applied to the EA task. Why didn't you consider this as a baseline or additional source of information in the framework?
> ```
>
> We first expain why no extra source of information is considered in this framework, then discuss the LLM-based method that exploits extra information and LLMs' knowledge, finally compare our work with LLM-based methods.
>
> ### **1.1 Why not consider additional information in the framework**
>
> In this work, we focus solely on structural information within our framework, as it represents a more general and widely applicable setting for the entity alignment task. In the EA field, structural signals are considered the **foundational resource** for alignment, enabling applicability even in low-resource or text-scarce scenarios.
>
>
>
> ### **1.2 Discussions of LLM-based entity alignment**
>
> While LLMs do offer significant potential for EA, especially in datasets rich with textual attributes. Recent studies have revealed the resource-intensive challenges in this research line. Specifically, (1) The **$O(|\mathcal{E}|^2)$** annotation space makes **LLM-API-based methods** costly; (2) The large parameter size of LLMs poses scalability limitations for **locally-deployed LLM-based methods**.
>
>
>
> ### **1.3 Comparison between NeuSymEA and LLM-based entity alignment**
>
> It is worth noting that our approach is **orthogonal** to LLM-based methods and can in fact **complement their limitations**. As shown in Table 2 of the manuscript, NeuSymEA is particularly effective in low-resource settings, making it well-suited to leverage LLM-generated pseudo-labels to improve alignment performance.
>
>
>
> To empirically support this, we compare NeuSymEA with a recently published LLM-based method [1] by
>
> (1) adopting the annotated labels they generated; and
>
> (2) the text attributes for lexical-based simlarity intialization in the symbolic component
>
> |                     | D-W-15K |       | D-Y-15K |       | EN-FR-15K |       | EN-DE-15K |       |
> | ------------------- | ------- | ----- | ------- | ----- | --------- | ----- | --------- | ----- |
> |                     | Hit@1   | MRR   | Hit@1   | MRR   | Hit@1     | MRR   | Hit@1     | MRR   |
> | LLM4EA [1]          | 0.742   | 0.810 | 0.891   | 0.926 | 0.875     | 0.909 | 0.977     | 0.983 |
> | NeuSymEA+labels     | 0.887   | 0.903 | 0.924   | 0.933 | 0.892     | 0.914 | 0.976     | 0.981 |
> | NeuSymEA+attributes | 0.920   | 0.943 | 0.985   | 0.989 | 0.981     | 0.987 | 0.970     | 0.980 |
>
> - [1] LLM4EA: Entity alignment with noisy annotations from large language models. NeurIPS, 2024.
>
>
>
> ---
>
> ##  **[Q2]**
> ```
> eq. 1 and 2: It's unclear how the confidence and probability is inferred in the symbolic setting.
> ```
>
>
> Equations (1) and (2) introduce the formulation of symbolic inference rules and alignment probabilities, and the subsequent sections (particularly Equations 9–11) explain how they are instantiated.
>
> #### **Eq. (1)** introduces rule templates:
>
> $$
> w_{p, p^{\prime}}:\left(e_j \equiv e_j^{\prime}\right) \wedge p\left(e_i, e_j\right) \wedge p^{\prime}\left(e_i^{\prime}, e_j^{\prime}\right) \Rightarrow\left(e_i \equiv e_i^{\prime}\right)
> $$
>
> - Here, $p$ and $p^{\prime}$ are relational paths in source and target KGs .
> - $w_{p, p^{\prime}}$ is the confidence of the rule, computed compositionally in Eq. (11).
>
> Essentially, this template formulates how a new alignment is inferred in a symbolic model: they have a pair of aligned neighbors, and they have a pair of similar relational paths to these neighbors. The confidence of this rule support is related to the similarity of this path pair.
>
>
>
> **Eq. (2) defines alignment probability**:
>
> Eq. (2) is a global modeling of the alignment probability, by aggregating all the rules in the format of Eq. (1):
> $$
> p_w\left(v\left(e, e^{\prime}\right) \mid G, G^{\prime}\right)=\text { aggregated support from symbolic rules }
> $$
>
> Practically, this is computed using Equation (9), which aggregates the confidence scores of all symbolic rules supporting a candidate pair $(e, e')$ via aligned neighbors.
>
>
> ---
> ##  **[Q3]**
> ```
> line 194: What exactly does the split mean? is it a split over entities or relations? how would it work if you apply something learned from one dataset to another?
> ```
>
>
> We first clarify the meaning of the split in the experimental setting, and discuss how framework generalize from one dataset to another.
>
> ### **3.1 Clarification of the split**
>
> The split refers to a split over aligned entity pairs, i.e., ground truth labels.
>
> - It is not a split over entities or relations individually, but over the alignment labels between entities from the two KGs.
> - For example, in DBP15K there are 15,000 ground-truth aligned entity pairs. A 3:7 split means 4,500 are used for training and 10,500 for testing.
>
>
>
> ### **3.2 Transferability across datasets**
>
> - The rules and relation patterns (e.g., $\eta(r), p s u b\left(r \subseteq r^{\prime}\right)$ ) are KG-specific, as they depend on the structure and semantics of relations in each dataset.
> - Therefore, the symbolic rule weights are not directly transferable between datasets with different schemas or relation vocabularies.
> - However, the general mechanism (logic deduction, confidence composition, rule mining process) is dataset-agnostic and can be applied across datasets. What's learned is specific to each dataset, but the framework generalizes.
>
>
> We appreciate you time and effort in reviewing our feedback. And we hope it addresses your concerns.

---

### Official Review · Reviewer_xG55 · 2025-06-29

**Clarity:** 3
**Significance:** 3
**Originality:** 2
**Rating:** 4
**Confidence:** 3

**Summary:**

This paper introduces NeuSymEA, a neuro-symbolic framework for entity alignment in knowledge graphs. The authors propose a unified model that leverages a variational EM algorithm to combine the strengths of neural and symbolic reasoning. The key ideas include modeling the joint probability of alignments in a Markov random field and using a novel logical deduction method to efficiently handle long-range rules. The experimental results on several benchmark datasets demonstrate the effectiveness and interpretability of the proposed approach.

**Questions:**

See the Weakness.

**Ethical Concerns:**

["NO or VERY MINOR ethics concerns only"]

**Final Justification:**

The paper is solid. I decide to maintain my positive score.

**Limitations:**

Yes

**Quality:**

3

**Strengths And Weaknesses:**

**Strengths:**

1.  The unification of neural and symbolic methods via a variational EM algorithm is elegant and effective. By optimizing a single joint probability distribution, the framework allows the neural (E-step) and symbolic (M-step) components to mutually enhance one another. This principled approach creates a more coherent and powerful model compared to methods that treat the components separately.

2.  The use of logical deduction to improve efficiency is a significant contribution. By decomposing long rules into unit-length sub-rules, the method avoids the exponential search space typically associated with symbolic reasoning. This makes the inference complexity linear with respect to rule length, allowing the model to leverage complex, multi-hop patterns in a scalable manner.

3.  The framework demonstrates excellent performance, achieving state-of-the-art results on benchmark datasets and showing impressive robustness in low-resource scenarios. A key strength is its built-in explainer, which provides rule-based justifications and confidence scores for its predictions. This combination of high accuracy and strong interpretability is a major advantage.

**Weaknesses:**

1.  The comparison could be strengthened by including more recent baselines. The latest model in the comparison is from 2023.

2.  The paper lacks a direct, head-to-head efficiency comparison with its baselines. While an analysis of NeuSymEA's own scalability and resource usage is provided, there is no data comparing its runtime, memory, or GPU consumption against key competitors. This information is essential for fully evaluating the practical performance trade-offs of the framework.

---

> ### Author Rebuttal · Authors · 2025-07-31
>
> ---
> ##  **[Q1]**
> ```
> The comparison could be strengthened by including more recent baselines. The latest model in the comparison is from 2023.
> ```
>
> Thank you for the constructive suggestion, we include two baselines, ASGEA[1] and TFP[2] in the last two years to strengthen the comparison. Results are below.
>
> Note that these two algorithms adopt the 3:7 (train: test) split of datasets, different from our experiment setting in the manuscript. Therefore, we align our experimental setting with theirs in the comparison below.
>
> |           | ZH-EN |       | JA-EN |       | FR-EN |       |
> | --------- | ----- | ----- | ----- | ----- | ----- | ----- |
> |           | Hit@1 | MRR   | Hit@1 | MRR   | Hit@1 | MRR   |
> | ASGEA [1] | 0.560 | 0.660 | 0.595 | 0.690 | 0.653 | 0.745 |
> | TFP [2]   | 0.816 | 0.868 | 0.812 | 0.868 | 0.853 | 0.899 |
> | NeuSymEA  | 0.835 | 0.870 | 0.829 | 0.868 | 0.872 | 0.905 |
>
>
>
> - [1] ASGEA: Exploiting Logic Rules from Align-Subgraphs for Entity Alignment, CoRR, 2024
> - [2] Rethinking Smoothness for Fast and Adaptable Entity Alignment Decoding, NAACL 2025
>
>
>
> ##  **[Q2]**
> ```
> The paper lacks a direct, head-to-head efficiency comparison with its baselines. While an analysis of NeuSymEA's own scalability and resource usage is provided, there is no data comparing its runtime, memory, or GPU consumption against key competitors. This information is essential for fully evaluating the practical performance trade-offs of the framework.
> ```
>
>
> Below, we compare the efficiency with strong baselines on the `fr-en` dataset. Here, we adopt the lightEA as our base neural model.
>
> |            | GPU Mem(/MB) | Mem(/MB) | Run Time(s) | Hit@1 accuracy |
> | ---------- | ------------ | -------- | ----------- | -------------- |
> | RREA       | 8903         | 2330     | 2605        | 0.797          |
> | Dual-AMN   | 9002         | 4009     | 96          | 0.793          |
> | LightEA    | 1328         | 3801     | 8           | 0.827          |
> | NeuSymEA-L | 1328         | 4153     | 101         | 0.858          |
>
> Results show:
>
> - **GPU Memory-consumption**: NeuSymEA's GPU memory usage is comparable to that of the base neural model it employs, enabling it to be as efficient as highly optimized models like LightEA.
> - **RAM memory consumption**: While memory usage depends on the neural model, the loaded dataset, and the symbolic model, NeuSymEA maintains acceptable overhead by avoiding the creation of large intermediate variables during processing.
>
> - Its **runtime** is slower than Dual-AMN and LightEA, but faster than RREA. The major reason is that neuro-symbolic reasoning generally has a symbolic reasoning process executed on CPUs, which is slower than the neural model that is executed on GPUs. We have enabled parallel computation and batch processing in our implementation of the symbolic model (Appendix A.2), so theoretically, it can be further accelerated by using a larger batch size and more parallel processes.
>
> Thank you again for your detailed review. We sincerely appreciate your constructive comments that help us improve this manuscript, and we will include these results in the final version.

---

> > ### Comment · Reviewer_xG55 · 2025-08-05
> >
> > Thank you for your reply. I decide to maintain my positive score.

---

> > > ### Author Response · Authors · 2025-08-05
> > > **Response to reviewer xG55**
> > >
> > > We are deeply grateful to the reviewer for the encouraging feedback. It truly means a lot to us that our clarifications have helped address your concerns. We will diligently reflect all the discussed points in the updated manuscripts, and we sincerely appreciate your confirmation of your positive rating. It's incredibly motivating for us. Thank you again for your constructive comments throughout the review process.

---

### Official Review · Reviewer_E7Dm · 2025-07-01

**Clarity:** 2
**Significance:** 2
**Originality:** 2
**Rating:** 4
**Confidence:** 2

**Summary:**

The paper proposes a variational EM method to align two sets of entities from two knowledge bases/graphs. The idea is to have two entity-matching score models: NN model parameterizing a score function, and symbolic model relying on rules. The two models are trained together using a variational EM method. The paper shows that the proposed approach outperforms several baselines (NN and symbolic) on several benchmarks.

**Questions:**

Please see above.

**Ethical Concerns:**

["NO or VERY MINOR ethics concerns only"]

**Final Justification:**

The authors' response addresses my concerns, especially two of them:
1. the authors showed that their approach is EM (rather than simply co-training), though employing some heuristics to make the approach work.
2. the authors showed that the comparison with PRASE if fair.

**Limitations:**

Yes

**Paper Formatting Concerns:**

No concerns

**Quality:**

2

**Strengths And Weaknesses:**

## Strengths
The paper shows strong results on several benchmarks.

## Weaknesses

* The paper is quite difficult to read, and especially to check the correctness of the EM. For instance, why optimizing (4) & (5) will have the same effect of minimizing the KL? Could the authors explain the intuition behind (9, 10, 11)? why do they make sense?

* Following the "heuristic" of selecting "positive" examples, and the trick for equation (4), in the E and M steps, I'm not pretty sure whether this is EM (could the authors prove that the log-likelihood of the observed data increases?). Instead, I think this is more like a co-training approach because:
   * in step E: the symbolic model annotate data for the NN model
   * in step M: the NN model annotate data for the symbolic

* In section 4.1, it's unclear to me how a new rule is created. Do we have to consider all possible combinations of unit rules?

* Line 217 "NeuSymEA unifies them under a joint probability objective." -- which "joint probability" is that?

* One baseline is PRASE. However, this baseline, from the original paper, is unsupervised learning. Then is it fair for comparison? Also, from the PRASE paper, different benchmarks are used. How well does the proposed approach perform on those benchmarks?

---

> ### Author Rebuttal · Authors · 2025-07-31
>
> We sincerely appreciate the time and effort the reviewer has invested in reviewing our paper and providing detailed comments. We address the concerns in the following.
>
> ---
> ##  **[Q1]**
> ```
> The paper is quite difficult to read, and especially to check the correctness of the EM. For instance, why optimizing (4) & (5) will have the same effect of minimizing the KL? Could the authors explain the intuition behind (9, 10, 11)? why do they make sense?
> ```
>
> We answer these two questions one by one.
>
> ### **1.1 Eq. (4) & (5)**
>
> Optimizing Eq. (4) &. (5) approximates minimizing the KL divergence between the variational distribution $q_\theta$ and the true posterior $p_w\left(v_H \mid v_O\right)$. This is because:
>
> - Equation (3) shows the evidence lower bound (ELBO) formulation:
>
> $$
> \log p_w\left(v_O\right) \geq E_{q\left(v_H\right)}\left[\log p_w\left(v_O, v_H\right)\right]-E_{q\left(v_H\right)}\left[\log q\left(v_H\right)\right]
> $$
>
> - Optimizing ELBO is equivalent to maximizing the log-likelihood (lower bounded by ELBO) while minimizing the KL divergence.
> - However, we points out that directly minimizing the KL is intractable, so we instead optimize a *reverse* KL via a sampling-based surrogate (Eq. (4)):
>
> $$
> \sum_{\left(e, e^{\prime}\right)} E_{p_w\left(v\left(e, e^{\prime}\right) \mid v_O\right)} q_\theta\left(v_H\right)
> $$
>
> - **Intuition**: To optimize the above objective, the symbolic model generates high-confidence pairs $\rightarrow$ these are treated as positives to supervise the neural model $q_\theta$. The neural model thus approximates the true posterior $p_w\left(v_H \mid v_O\right)$, closing the KL gap indirectly. Equation (5) simply adds supervision from observed data to stabilize training.
>
> ### **1.2 Eq. (9) & (10) & (11)**
>
> Allow us to clarify any confusion. These three equations describe the symbolic update process using decomposed rules:
>
>  **Eq. (9): Inference via neighbor aggregation**
>
> - This computes the alignment probability $p_w\left(v\left(e, e^{\prime}\right)\right)$ by aggregating support from neighboring aligned pairs $\left(e_t, e_t^{\prime}\right)$ via one-step (unit-length) rules.
> - Each neighbor contributes a confidence score based on:
>
>   - $\eta(r)$ : the relation pattern of $r$, quantifying its uniqueness (e.g., 1-to-1 vs many-to-many).
>
>   - $p_{\text {sub }}\left(r \subseteq r^{\prime}\right)$ : the subrelation probability, i.e., how likely $r$ is structurally or semantically a subrelation of $r^{\prime}$.
>
>
> - The product form ensures that support from multiple neighbors is combined in a noisy-OR fashion: as more confident neighbors exist, the overall alignment probability increases.
>
> This equation reflects a forward chaining deduction, where alignments propagate from neighbors under weighted logical rules.
>
> **Eq. (10): Updating subrelation probabilities**
>
> - This equation updates the confidence that $r \subseteq r^{\prime}$ holds across the two KGs.
> - It does so by measuring how often aligned entity pairs co-occur under matching relation pairs.
>
> The result is a data-driven estimation of relation-level correspondence across KGs.
>
> **Eq. (11): Rule confidence via composition**
>
> - This computes the final confidence score of a rule $p, p^{\prime}$ of length $L$ by multiplying the confidences of each unit step.
>
>
> ---
> ##  **[Q2]**
>
> ```
> Following the "heuristic" of selecting "positive" examples, and the trick for equation (4), in the E and M steps, I'm not pretty sure whether this is EM (could the authors prove that the log-likelihood of the observed data increases?). Instead, I think this is more like a co-training approach...
> ```
>
> Thank you for your insightful comments, and allow us to clarify any confusion. While our E/M steps may resemble co-training heuristics in form, our method is theoretically grounded in variational EM.
>
> Specifically, we formalize a latent-variable model where the truth scores $v_H$ are hidden variables, and model the joint probability $p_w(v_O, v_H)$ over observed and hidden variables(truth scores). Our E-step updates the variational distribution $q_\theta$ to approximate the true posterior, and the M-step maximizes the expected complete log-likelihood with respect to $w$, given $q_\theta$. This alternating optimization provably increases the ELBO (a lower bound of the true likelihood), which satisfies the formal criteria of EM.
>
> I acknowledge that the use of heuristics such as greedy one-to-one filtering in the E-step may blur the boundary. But these heuristics are used only for practical implementation of pseudo-labeling and do not violate the variational EM principle, since the neural model is still trained to minimize divergence from symbolic inference. Fig2 shows proxy metrics (neural MRR, symbolic precision) improve over iterations.
>
>
> ---
> ##  **[Q3]**
> ```
> In section 4.1, it's unclear to me how a new rule is created. Do we have to consider all possible combinations of unit rules?
> ```
>
> The rule creation process is clarified in Equations (7)-(8):
>
> - A long rule of length $L$ is decomposed into a conjunction of $L$ unit-length subrules.
> - During optimization, only unit-length rules are explicitly learned, and longer rules are constructed by deductive composition.
>
> The symbolic model doesn't consider all combinations exhaustively. Instead, we exploit the local sparsity of KGs to mine rules efficiently:
>
> - For each candidate alignment ( $e, e^{\prime}$ ), it uses breadth-first search to find matching paths of length $\leq$ $L$.
> - Each matched pair of paths forms a potential rule $p, p^{\prime}$, and its confidence is computed using Equation (11).
>
> Thus, the search is constrained to reachable anchor pairs and their paths, avoiding exhaustive enumeration. As a result, the number of rules mined is a small number (as shown in middle & right in Fig. 4).
>
> ---
> ##  **[Q4]**
> ```
> Line 217 "NeuSymEA unifies them under a joint probability objective." -- which "joint probability" is that?
> ```
>
> This joint probability refers to:
>
> $
> p_w\left(v_O, v_H\right)
> $
>
> modeled via a Markov Random Field as described in Section 3:
>
> - The Markov Random Field captures interactions among all entity pairs via symbolic rules.
> - This joint distribution is optimized by variational EM (Equation 3), making it the central objective function unifying symbolic and neural inference.
>
> This "joint probability" represents the coherent agreement between symbolic rule-based reasoning and neural embeddings, governed by shared latent variables $v_H$.
>
>
> ---
> ##  **[Q5]**
> ```
> One baseline is PRASE. However, this baseline, from the original paper, is unsupervised learning. Then is it fair for comparison? Also, from the PRASE paper, different benchmarks are used. How well does the proposed approach perform on those benchmarks?
> ```
>
> We respectfully clarify the **fairness** of the experimental setting by comparing our methods under both their "unsupervised" setting and our supervised setting.
>
> ### **5.1 Unsupervised setting with text attributes for alignment.**
>
> Different from our setting that only structure information is used, the "unsupervised" setting in PRASE uses text attributes as **extra information**. Specifically, they treat attribute values as entities and exploit the same attribute values as seed alignments to guide the entity alignment.
>
> NeuSymEA can also be run by constructing attributed KGs, and enable unsupervised learning. We attach the hit@1 performance under this setting below, on the benchmark datasets they adopted (D-W-15K, D-W-100K, D-Y-100K, EN-FR-100K, EN-DE-100K), for comprehensive comparison.
>
> | Hit@1    | D-W-15K | D-W-100K | D-Y-100K | EN-FR-100K | EN-DE-100K |
> | -------- | ------- | -------- | -------- | ---------- | ---------- |
> | PRASE    | 0.875   | 0.804    | 0.993    | 0.930      | 0.955      |
> | NeuSymEA | 0.920   | 0.845    | 0.994    | 0.942      | 0.956      |
>
>
>
> ### **5.2 Our experimental setting enable fair comparison by supervised learning of PRASE with same input.**
>
> Our setting is a more general setting where structure information is the main information and no attributes are available. In this case, PRASE utilize the **same input** (two KGs, and a set of seed alignments)  as NeuSymEA for supervised learning, rather than unsupervised, ensuring fair comparison.
>
> | Hit@1    | JA-EN | FR-EN | ZH-EN |
> | -------- | ----- | ----- | ----- |
> | PRASE    | 0.611 | 0.647 | 0.652 |
> | NeuSymEA | 0.811 | 0.858 | 0.804 |
>
>
>
> Thank you for taking the time to review our response. We hope it has clarified any confusion and addressed your concerns.

---

> > ### Comment · Reviewer_E7Dm · 2025-08-05
> >
> > I would like to thank the authors for the response, which addresses my concerns. I raise the score.

---

> ### Author Response · Authors · 2025-08-05
> **Response to reviewer E7Dm**
>
> Dear reviewer E7Dm,
>
> We are glad that our work has been acknowledged by you following the discussions. We sincerely thank you for the time and effort you invested in reviewing our work and engaging in the discussion. Your detailed and professional comments are invaluable for improving our work, and we will integrate all discussed points into the revised manuscript. Thank you once again for your support.

---

### Official Review · Reviewer_J4kQ · 2025-07-01

**Clarity:** 3
**Significance:** 3
**Originality:** 2
**Rating:** 4
**Confidence:** 2

**Summary:**

This research introduces NeuSymEA, a system for entity alignment between two knowledge graphs. The system combines symbolic reasoning (which is accurate and interpretable) with neural models (which handle large-scale noisy data well).
Previous work focused on single knowledge graphs, but NeuSymEA specifically handles alignment across different knowledge graphs (currently no more than two at a time).
The system treats alignment as a joint probability problem using a Markov random field with logical rule guidance. It uses a variational EM algorithm for training: the neural model predicts alignment probabilities (E-step), then the symbolic model updates rule weights (M-step). To speed up inference, long logic rules are broken into smaller steps.
The system also includes an explainer module that shows which logic paths support each predicted alignment. NeuSymEA achieves strong performance on benchmark datasets, especially with limited labeled data, demonstrating both effectiveness and interpretability.

**Questions:**

1) In M-step description the paper states that the model uses a greedy one-to-one matching strategy to reduce false positives during pseudo-label generation. However, this step appears to be a heuristic and is not rigorously analyzed. Could you elaborate on how NeuSymEA's performance is affected by this heuristic and which alternative approaches have you considered?

2) I understand that the authors mentioned this following as limitation of their current work. Still, how do you envision NeuSymEA extending to multi-KG scenario? Could you briefly describe how such an extension could be designed and what would be the pain points?

**Ethical Concerns:**

["NO or VERY MINOR ethics concerns only"]

**Limitations:**

The main limitation of this work (as mentioned by the authors) the fact that NeuSymEA is designed for pairwise alignment only and multi-KG cases are currently not supported.

**Paper Formatting Concerns:**

No formatting concerns.

**Quality:**

3

**Strengths And Weaknesses:**

Strengths

Quality

The proposed method is technically sound and well-motivated. The use of a variational EM algorithm to jointly optimize symbolic rule weights and neural parameters is principled and well-grounded in probabilistic modeling.
The paper provides a clever decomposition strategy for long logic rules, which significantly improves inference scalability without compromising symbolic reasoning depth.
Empirical evaluation is comprehensive and includes standard EA benchmarks (e.g., DBP15K) with both high-resource and low-resource settings, showcasing consistent gains over prior baselines.

Clarity
The overall structure is logical, with motivation, methodology, and experiments clearly stated.
Key components such as the symbolic explainer, logical decomposition, and variational EM training are explained with sufficient technical detail.
Despite the technical complexity, the authors make a commendable effort to keep the exposition readable, especially around the symbolic-neural interaction.

Significance
This paper tackles a key challenge: matching information across different knowledge sources, not just fill in missing facts within one.
The proposed method improves both accuracy and explainability—two things that usually don’t go well together in ML systems.
It also includes a feature (explainer component) that explains why the system made a match, which is especially useful in fields like medicine or language translation, where understanding the reasoning behind results is very important.

Originality

This work introduces a new way to combine logic rules with a probabilistic learning method to match entities across different knowledge graphs.
It also includes a smart trick: breaking down long and complex logic rules into smaller, simple steps to keep the system fast and manageable.
Finally, it adds an explanation tool that works in two modes—one using only known matches, and another using both known and predicted matches—making the system’s decisions easier to understand and trust.


Weaknesses
Quality

The paper should better explain how inference quality is affected by pseudo-label noise in the early stages.
The one-to-one matching heuristic used to reduce false positives is not rigorously analyzed or justified — a more formal treatment or ablation would strengthen the argument.

Clarity

Certain parts of the paper, especially those introducing dense notation (e.g., η, p_sub, rule decomposition), could be more clearly explained or accompanied by examples. In particular, equations sometimes appear without adequate narrative grounding.
The treatment of evaluation metrics across symbolic and neural models is briefly reconciled, but the equivalence assumption between recall and hit@1 could be more carefully justified.

Significance

Right now, the method can only match two knowledge graphs at a time, and the authors don’t yet explain how it could be expanded to handle many at once.
It’s also not clear how well the system works when the graphs are very different in structure or topic, which would be useful to test.
While the paper presents the idea of combining logic and learning as something new, some parts seem borrowed from earlier work. It would help if the authors made it clearer what parts are truly original.

---

> ### Author Rebuttal · Authors · 2025-07-31
>
> We appreciate the detailed comments from reviewer, and address the two questions below:
>
> ##  **[Q1]**
>
> ```
> Question: In M-step description the paper states that the model uses a greedy one-to-one matching strategy to reduce false positives during pseudo-label generation. However, this step appears to be a heuristic and is not rigorously analyzed. Could you elaborate on how NeuSymEA's performance is affected by this heuristic and which alternative approaches have you considered?
> ```
>
>
> Thank you for your question, we would like to take this oppotunity to present the details here. Below, we first list two alternative approaches we have considered, then we show how our heuristic reduce false positives by showing the truth positive rate, finally show how the NeuSymEA's alignment performance is affected by this heuristic.
>
>
>
> ### **1.1 Alternative approaches**
>
> We include two algorithms: (1) Threshold: select entity pairs by using a preset threshold to select pairs with high matching scores; (2) Top1: each candidate entity along with its top1 matched entity is added to the pseudo-label set.
>
> The shortage of these alternative solutions are:
>
> - The thresholding method is sensitive to the preset threshold, introducing burden to the hyperparameter tuning.
> - Both these algorithms can potentially generate conflict pairs (identical head or tail entities in different entity pairs).
>
> The greedy one-on-one matching is essentially a top1 strategy that naturally avoids conflit pairs.
>
>
>
> ### **1.2 The true positve rate (TPR) of pseudo-labels is increased by the one-on-one matching**
>
> Taking the `fr-en` dataset as an example, the true postive rate in the 5 steps are below.
>
> | Step                | 1     | 2     | 3     | 4     | 5     |
> | ------------------- | ----- | ----- | ----- | ----- | ----- |
> | Thresholding        | 0.452 | 0.604 | 0.639 | 0.655 | 0.654 |
> | Top1                | 0.731 | 0.826 | 0.839 | 0.847 | 0.848 |
> | One-on-One matching | 0.748 | 0.839 | 0.859 | 0.860 | 0.860 |
>
> We can observe a significant improvement of truth postive rate by the one-on-one matching algorithm. And we also observe that the TPR of pseudo labels grows as the EM-steps process.
>
>
>
> ### **1.3 Impact of the strategies on NeuSymEA's final Perofrmance**
>
> |                     | Hit@1     | MRR       |
> | ------------------- | --------- | --------- |
> | Thresholding        | 0.836     | 0.875     |
> | Top1                | 0.848     | 0.884     |
> | One-on-One matching | **0.858** | **0.894** |
>
> Results reveal that: One-on-one matching enhance the final performance with its precise pseudo-labels.
>
> ---
>
> ##  **[Q2]**
> ```
> Question: I understand that the authors mentioned this following as limitation of their current work. Still, how do you envision NeuSymEA extending to multi-KG scenario? Could you briefly describe how such an extension could be designed and what would be the pain points?
> ```
>
> We envision extending NeuSymEA to multi-KG scenario with two directions of design: (1) recursive pair-wise KG alignment; and (2) simultaneous multi-KG alignment in a unified space. Below we detail the two designs and their painpoints.
>
> ### **2.1 Recursive pair-wise alignment**
>
> The first design is naive. Simply apply the current NeuSymEA framework to perform entity alignment between every KG pair, finally recursively merge these multiple KGs.
>
> **The painpoints in the first design** are: (1) The number of KG pairs grows quadratically to the number of KGs, presenting efficiency and effectiveness challenges in scheduling the recursion. (2) Conflicts among alignments. For instance, the model infers three alignment pairs $(e^1_i=e^2_i), (e^2_i= e^3_i), (e^3_i= e^1_j)\text{ for three KGs } G^1, G^2, G^3$, there is an conflict if $i\neq j$.
>
>
>
> ### **2.2 Simultaneous multi-KG alignment**
>
> The second direction is more ambitious and theoretically appealing to jointly align all KGs in a unified probabilistic framework. This requires several key modifications:
>
> **Modification 1.** Generalized alignment variables
>
> Instead of binary variables $v\left(e, e^{\prime}\right) \in\{0,1\}$, we would define alignment clusters: each entity belongs to one equivalence class (i.e., same real-world entity). This leads to a clustering-based formulation rather than pairwise binary classification.
>
> **Modification 2.** Multi-source symbolic rule inference:
>
> Current rules involve relational paths in two KGs (e.g., $p, p^{\prime}$ ). In multi-KG settings, we must reason over triplets of paths (or more), i.e., $p_{i j}, p_{j k}, p_{i k}$ across $G_i, G_j, G_k$ and respect transitive consistency: i.e., if $e_i \equiv e_j$ and $e_j \equiv e_k$, then $e_i \equiv e_k$.
>
> **Modification 3.** Joint probabilistic model across all KGs:
>
> - Extend the Markov random field to include higher-order potentials that model transitive consistency among multiple KGs.
> - The neural model would also need to be extended to learn shared embedding space across all KGs.
>
> **The painpoint in the second direction** is that: the increased heterogeneity in multiple KGs, challenging the robustness of the extended framework.
>
>
>
> Above is how we envision extending NeuSymEA to multi-KG scenario, in terms of both the potential design and painpoints. We hope this vision can inspire future research on scalable and robust multi-KG alignment.

---

> > ### Comment · Area_Chair_VqqC · 2025-08-09
> > **Official Comment by Area Chair**
> >
> > Dear J4kQ,
> >
> > We'd love to hear your thoughts on the rebuttal. If the authors have addressed your rebuttal questions, please let them know. If the authors have not addressed your rebuttal questions, please inform them accordingly.
> >
> > Thanks, Your AC

---

### Decision · Program_Chairs · 2025-09-17

**Decision:**

Accept (poster)

**Comment:**

NeuSymEA presents a unified neuro-symbolic framework for pairwise entity alignment (EA) between knowledge graphs.  It models the joint probability of all candidate alignments via a Markov random field whose potentials are defined by first-order logical rules. The paper unifies historically disjoint neural and symbolic EA communities in a single, theoretically grounded framework, achieving accuracy, robustness, and interpretability simultaneously—an important milestone for trustworthy knowledge-graph integration.